# Thioredoxin System in Insects: Uncovering the Roles of Thioredoxins and Thioredoxin Reductase beyond the Antioxidant Defences

**DOI:** 10.3390/insects15100797

**Published:** 2024-10-14

**Authors:** Andrea Gřešková, Marek Petřivalský

**Affiliations:** Department of Biochemistry, Faculty of Science, Palacký University in Olomouc, Šlechtitelů 27, 77900 Olomouc, Czech Republic

**Keywords:** antioxidant system, *Drosophila melanogaster*, honey bee, reactive oxygen species, redox signalling, thioredoxin, thioredoxin reductase

## Abstract

**Simple Summary:**

The utilisation of oxygen for cellular respiration and other metabolic pathways in aerobic organisms is inherently associated with the production of reactive oxygen species, a group of highly reactive and oxidising small molecules. Reactive oxygen species can cause oxidative modifications of biomolecules and cellular components, thus compromising their native structure and biological functions. The role of the antioxidant defence is to tightly control levels of reactive species by connecting diverse antioxidant small molecules and antioxidant enzymes. This review summarises the actual knowledge on the role of the crucial part of the antioxidant system in insects termed the thioredoxin system. It is composed of redox-active protein thioredoxin, which interacts with and reduces the oxidised forms of multiple proteins. It is assisted by the enzyme thioredoxin reductase, catalysing the reductive regeneration of thioredoxins. Recent advances have uncovered how the functions of the thioredoxin system extend beyond its primary function in controlling the cell redox state to diverse developmental processes and the regulation of stress and immune responses.

**Abstract:**

Increased levels of reactive oxygen species (ROS) produced during aerobic metabolism in animals can negatively affect the intracellular redox status, cause oxidative stress and interfere with physiological processes in the cells. The antioxidant defence regulates ROS levels by interplaying diverse enzymes and non-enzymatic metabolites. The thioredoxin system, consisting of the enzyme thioredoxin reductase (TrxR), the redox-active protein thioredoxin (Trx) and NADPH, represent a crucial component of antioxidant defence. It is involved in the signalling and regulation of multiple developmental processes, such as cell proliferation or apoptotic death. Insects have evolved unique variations of TrxR, which resemble mammalian enzymes in overall structure and catalytic mechanisms, but the selenocysteine–cysteine pair in the active site is replaced by a cysteine–cysteine pair typical of bacteria. Moreover, the role of the thioredoxin system in insects is indispensable due to the absence of glutathione reductase, an essential enzyme of the glutathione system. However, the functions of the Trx system in insects are still poorly characterised. In the present review, we provide a critical overview of the current knowledge on the insect Trx system, focusing mainly on TrxR’s role in the antioxidant and immune system of model insect species.

## 1. Introduction

The antioxidant system has evolved as a regulatory and protective mechanism to cope with reactive oxygen (ROS) and reactive nitrogen (RNS) species, which are continuously generated within the cells and tissues of aerobic organisms. These reactive species are by-products of aerobic metabolism but are also produced by specific metabolic pathways and enzymes, and are involved in cell redox signalling, developmental processes and immune defence [1,2]. Various enzymatic and non-enzymatic mechanisms regulate the ROS and RNS levels and thus play an indispensable role in the proper functioning of aerobic organisms [3]. In an intracellular steady state, ROS production and removal are in a balanced equilibrium specific to each organelle [4,5,6]. For redox signalling to occur, this equilibrium must be disturbed locally in a time- and site-specific manner, either by increasing the ROS production or by partially decreasing the activity of the antioxidant systems. Under physiological conditions, this process is regulated, but under oxidative stress, antioxidant defences are insufficient to control higher ROS levels [7,8,9,10]. ROS or RNS levels elevated above the capacity of their removal and catabolism result in oxidative or nitro-oxidative stress, respectively, associated with the modification of biomolecules and damage to major cellular components. Lipids, proteins and nucleic acids can be oxidised or undergo other modifications, which interfere with their biological functions. The subsequent destructive mechanisms initiated by highly increased ROS or RNS levels can contribute to ageing, carcinogenesis and cell death [11,12,13].

In proteins, certain amino acids are highly sensitive to oxidative and nitrosative modifications, with cysteine being particularly important. The thiol group of cysteine can undergo various reversible and irreversible oxidative modifications that affect the structure and function of the proteins, which subsequently leads to changes in the activities of metabolic enzymes, transcription factors and components of membranes [14,15]. The thioredoxin system consists of thioredoxins (Trxs) as specific redox-active proteins containing functional thiol groups and the enzyme thioredoxin reductase (TrxR), which catalyses the reduction of oxidised thioredoxins utilising NADPH as the reductive cofactor. The components of the Trx system act in dithiol disulphide exchange reactions to regulate the metabolic pathways of the cells. They primarily serve as electron donors for antioxidant system enzymes and regulate proteins in response to changing redox environments [16,17,18,19].

Similar to other aerobic organisms, the Trx system represents a key component of the insect’s antioxidant system. Moreover, the role of the Trx system in insects is indispensable because, unlike other organisms, insects lack glutathione reductase, the crucial enzyme of the glutathione system, another critical part of cellular redox regulation [20,21]. In addition to antioxidant protection, Trxs are involved in signalling, proliferation and apoptosis in insect cells and tissues [22]. Interestingly, insects have evolved unique variations of thioredoxin reductases bearing features of higher eukaryotes and simpler prokaryotes. Insect TrxRs resemble mammalian enzymes in the overall structure and mechanism of action. Still, the selenocysteine–cysteine pair in the active site is replaced by a Cys-Cys pair typical of bacteria.

In this review, we present an overview and critical assessment of the current knowledge of the insect Trx system and the structure and molecular properties of its components. The indispensable functions of the Trx system in insect developmental and growth processes and stress responses are highlighted. Besides the accumulated knowledge obtained in studies of model insect species like *Drosophila* and *Anopheles* spp., we also focus on the honey bee (*Apis mellifera*), a representative of social insects and an important pollinator. Understanding the role of the Trxs system within the antioxidant mechanisms of bees can contribute to finding strategies for bee protection from stress conditions associated with malnutrition, pathogen infections or pesticide exposures.

## 2. Brief Overview of Insect Antioxidant System

### 2.1. Reactive Oxygen Species and Oxidative Stress in Insects

ROS are highly reactive chemical molecules, namely free radicals containing unpaired electrons or non-radical molecules formed from oxygen during the oxidation of other electron-carrying molecules [23]. Superoxide anion radical, hydrogen peroxide, hydroxyl radical, singlet oxygen and hypochlorous acid are the most relevant ROS produced in animal cells [24]. On the quantitative scale, the production of reactive species occurs mainly in the mitochondrial inner membrane, where complexes I and III of the electron transport chain are the primary sources of superoxide and hydroxyl radicals. Peroxisomes, plasma membrane NADPH-oxidases (EC 1.6.3.1), cytosolic xanthine oxidases (EC 1.17.3.2) and cytochromes P450 also belong to important ROS producers. ROS regulate various cellular mechanisms, such as signalling, senescence and apoptosis, or may act as antimicrobial agents within insect immune responses [7,8,25].

The reaction of ROS with DNA results in the oxidation of nitrogenous bases and deoxyribose, followed by DNA fragmentation and disrupted DNA condensation and chromatin coiling. DNA oxidation can also lead to base substitutions or double helix breaks [26]. Cell membranes are susceptible to radical damage due to the reactivity of polyunsaturated fatty acids. The unsaturated chains in membrane phospholipids are peroxidised by ROS to form hydroperoxide and alkyl radicals, triggering a chain reaction and peroxidation of other lipids. This process results in a change in the structure of the membrane, affecting its fluidity and damaging its integrity [27]. ROS affect proteins and enzymes by oxidising amino acid residues or cleaving peptide bonds, interfering with their catalytic activity or regulatory functions [25].

The level of oxidative stress in insect organisms is not constant but varies depending on developmental stage, environmental conditions and activity level [28,29]. Insects are burdened by oxidative stress due to their lifestyle. The respiratory system of insects consists of tracheal tubular tubes that distribute high amounts of oxygen directly to the tissues, where it diffuses to intracellular spaces and mitochondria. Furthermore, oxygen consumption in insect tissues is highly increased during demanding physical activities such as in thoracic muscles during flight [30]. Adult flying insects are more susceptible to oxidative stress than their larval stages or non-flying species. Increased ROS production is also known to occur during the interactions of the insect gut epithelium with pathogenic or symbiotic microbes or during the so-called oxidative burst of haemocytes in response to invading microbial pathogens. Environmental pollutants (heavy metals, insecticides, herbicides, etc.) and diverse plant-derived compounds forming part of plant defences against herbivores induce oxidative stress [3,21].

Blood-sucking insects are at greater risk of oxidative stress due to iron released from haemoglobin molecules, which can participate in chemical reactions that produce highly reactive ROS. In the Fenton reaction, free ferrous ions catalyse the conversion of hydrogen peroxide to a hydroxyl radical, which can trigger a cascade of damaging reactions, including membrane lipid peroxidation and DNA base oxidation [25].

Antioxidants are substances able, even at relatively low concentrations, to efficiently react with or catalytically decompose oxidising compounds and, thus, significantly slow or inhibit the oxidation of biomolecules, which are the targets of pro-oxidants [31]. ROS levels and the redox homeostasis of cells, tissues and the whole organism are controlled by multiple synergistic and overlapping enzymatic and non-enzymatic antioxidant components. Like other animals, several ROS-scavenging enzymes operate in insects with diverse isoforms showing a specific localisation in the cytosol or cell organelles. Superoxide dismutases (SOD, EC 1.15.1.1) catalyse the conversion of the superoxide anion radical to hydrogen peroxide, which can be converted to water by the action of catalase (CAT, EC 1.11.1.6) or peroxidases (POX, EC 1.11.1). Both SOD and CAT serve as universal antioxidant enzymes in most aerobic forms of life, including insects, but some types of peroxidases have not been detected in all species examined. Specifically, glutathione peroxidase (GPx, EC 1.11.1.9), which can reduce both inorganic H_2_O_2_ and organic lipid peroxides, is not present in insects. However, the glutathione-peroxidase activity to catalyse the reduction of lipid peroxides is provided by specific isoforms of glutathione transferases (GST, EC 2.5.1.18) [21,32].

A chemically diverse group of non-enzymatic substances, including carotenoids, tocopherols, ascorbic acid, glutathione (GSH) and thioredoxins, is involved in antioxidant protection [4,11,33]. The regeneration of the reduced form of these antioxidants is accomplished using NADH for ascorbic acid, NADPH-dependent glutathione reductase (GR, EC 1.8.1.7) for glutathione and NADPH-dependent thioredoxin reductase (TrxR, EC 1.8.1.9) for thioredoxin [32,34]. However, GR is absent in all the insect species studied so far. GSH reduction is mediated by the thioredoxin system, which thus plays a vital role in GSH-dependent processes beyond its antioxidant role, i.e., in xenobiotic detoxification (Figure 1) [21,35]. Other essential antioxidants active both in intracellular compartments and the extracellular haemolymph include specific carbohydrates, polyols, uric acid and protein vitellogenin [3,36,37].

### 2.2. Non-Enzyme Components of Insect Antioxidant System

Tocopherols and tocotrienols are lipophilic fat-soluble substances with an important role in the protection of membrane lipids from oxidative damage [38]. Tocopherols scavenge fatty acid peroxyl radicals and thus prevent the spreading of peroxidative chain reactions within membranes. The primary function of vitamin E in model insect *D. melanogaster* is to support the antioxidant system under stressful conditions [39,40]. L-ascorbic acid is a water-soluble substance that acts as an antioxidant or enzyme cofactor. Most animals, including insects, cannot synthesise it and must ingest ascorbate in their plant-based diet components [41]. Glutathione, a tripeptide consisting of glutamate, cysteine and glycine, occurs in high intracellular levels mainly as the reduced (GSH) and also oxidised (GSSG) form, produced by joining two GSH molecules via a disulphide bridge. GSH is involved in diverse metabolic and antioxidant pathways as an enzyme cofactor, but it functions as an important antioxidant directly reacting with ROS. GSSG can be converted back to its reduced form by the action of GR and NADPH. However, as insect species have lost GR genes during evolution, the thioredoxin system provides the reductive glutathione regeneration function instead. In this case, GSSG is reduced to GSH by the reduced Trx protein, and then the oxidised Trx is reduced back by the enzyme thioredoxin reductase in the presence of NADPH [35,41]. Carotenoids are a group of isoprenoid compounds containing up to 15 conjugated double bonds in their structure, with β-carotene as the most crucial representative [42]. Carotenoids are synthesised by some bacteria, fungi and plants, which can serve as a food source of carotenoids for insects. They are essential for multiple biological functions of insects, including vision, reproduction and immune responses. For bees, pollen is an essential source of α- and β-carotene [43].

Transferrin and ferritin are proteins that transport and store iron in insects to prevent its harmful effects [44,45]. Insect transferrins also act as vitellogenic proteins in the egg during yolk formation. In addition, the amount of transferrin increases in the stress response to ingested toxic compounds, injury and light [44,45]. The main function of ferritin is iron transport and storage, mainly in insect eggs [46,47]. In *D. melanogaster*, genes encoding ferritin are significantly upregulated when exposed to oxidative stress, suggesting an antioxidant role for these proteins. The antioxidant functions of these proteins are also suggested by their increased levels following *A. aegypti* blood feeding, related to their involvement in removing excess iron from digested blood meal [44,48]. Vitellogenin protein is found in egg-laying animals such as insects, fish and birds, and it has a primary role as a precursor of yolk proteins. In insects, in addition to reproductive processes, it is involved in immune defence, increasing the lifespan and antioxidant protection. Higher levels of vitellogenin have been reported in queens and the winter generation of worker bees, where it is associated with greater resistance to oxidative stress and, thus, an increased lifespan. The antioxidant function of vitellogenin has been suggested in the fat body of both *A. mellifera* and *A. cerana* adult worker bees [37,49].

### 2.3. Enzyme Components of Insect Antioxidant System

Antioxidant enzymes are divided into primary and secondary groups according to their function in the structured hierarchy of interconnected antioxidant systems. Primary enzymes directly catalyse ROS conversion and include superoxide dismutase (SOD), catalase and peroxidases. Secondary enzymes transfer electrons from reduced cofactors to oxidised components, including thioredoxin reductases, glutathione reductases and methionine sulphoxide reductases [5,21,50].

Superoxide dismutases (SODs) catalyse the dismutation of superoxide anion radicals to oxygen and hydrogen peroxide. SODs in animals are found in two forms differing in cellular localisation and active site structure—Cu/ZnSOD (SOD1) localise in the cytoplasm and MnSOD (SOD2) in the mitochondrial matrix. Moreover, extracellular Cu/ZnSODs have been identified in many animal and insect species [21]. The family of Cu/ZnSODs in *D. melanogaster* include genes for cytoplasmic Cu/ZnSOD, extracellular SOD (Sod3), copper-containing chaperone (CCS) and two genes structurally related to Sod with an uncharacterised function (Rsod and Sodq) [51]. Studies in insects and animals suggested that increased SOD activity decreases oxidative damage in the cells and might also contribute to an increased life span [52,53]. However, other studies do not support a straightforward correlation between SOD activity and decelerated ageing in insects [51,54]. Catalase (CAT) catalyses the conversion of two molecules of hydrogen peroxide into water and oxygen, preventing the formation of hydroxyl radicals [5,21]. However, due to its high Km value for peroxide, CAT is less efficient at removing H_2_O_2_ at lower intracellular levels than peroxidases [30]. Catalase is typically enriched in the matrix of peroxisomes, organelles with the highest levels of H_2_O_2_ in animal and insect cells [55]. 

Glutathione S-transferases (GSTs) are a large family of multifunctional enzymes classified based on their location in the cell into cytosolic and microsomal [56,57]. The main function of GSTs is the detoxification of electrophilic xenobiotic compounds by their conjugation with a nucleophilic thiol moiety of reduced GSH. GSTs also play a role in intracellular transport, hormone biosynthesis and protection against oxidative stress, specifically peroxidative damage [21,56]. A total of 40, 35 and 11 cytosolic GST isoforms have been identified in *D. melanogaster*, *A. gambiae* and *A. mellifera*, respectively. In contrast, only three microsomal GSTs have been identified in *D. melanogaster*, three in *A. gambiae* and two in *A. mellifera* [57,58]. In particular, sigma class of GSTs have an important role in antioxidant defence. The expansion of the sigma class GSTs in honey bees is likely related to protection against ROS produced by highly intensive aerobic metabolism [21,59].

Glutathione peroxidases (GPx) are a group of enzymes that catalyse the reduction of hydrogen peroxide or organic hydroperoxides to water or their respective alcohols using reduced glutathione and thus perform an essential function in the antioxidant system [60,61,62]. GRs catalyse the reduction of GSSG to GSH by maintaining high intracellular levels of reduced glutathione, which is crucial for cellular redox balance and regulation. However, genes encoding GPXs and GRs were lost in insects [21,35]. Two homologous genes with thioredoxin reductase (GPTx) activity have been described in *A. mellifera*, *A. gambiae*, and *D. melanogaster*. These enzymes show a higher affinity for thioredoxins used as an electron donor instead of glutathione. The specificity of GPX/GRs for thioredoxin or glutathione appears to be related to the structure of the enzyme active site. Glutathione peroxidases containing selenocysteine in the active site have greater specificity for glutathione, whereas cysteine-containing enzymes prefer thioredoxin as an electron donor. Insect GPTx structures contain two redox centres with cysteine residues that form a disulphide bond during catalysis [61].

Ascorbate peroxidases (APx) are enzymes required to reduce H_2_O_2_ using ascorbic acid as an electron donor [36]. In insects, in contrast to plants, the presence of APx at the gene or protein level has not yet been demonstrated; therefore, other peroxidases, enzymes known for their generally broader substrate specificity, might exhibit the observed Asc-dependent peroxidase activities in insect cells and extracts. APXs, due to their low Km value for H_2_O_2_, are effective in removing low peroxide concentrations that catalases cannot scavenge [63].

Thioredoxin peroxidases (TPx), also called peroxiredoxins, are a family of cysteine-based peroxidases that eliminate H_2_O_2_ and organic hydroperoxides [64]. TPx catalyses the reduction of H_2_O_2_ to H_2_O using reduced Trxs. These reactions affect both H_2_O_2_-mediated redox signalling and Trx interactions with other target molecules [16]. Based on the number of active Cys residues in their protein sequences, TPxs are classified into two subfamilies: 1-Cys and 2-Cys. Unlike 1-Cys enzymes, 2-Cys proteins have an additional Cys residue in the C-terminal region. The 2-Cys group is subdivided into typical and atypical enzymes based on structural features and biochemical properties [64]. Five TPx genes have been identified in the *D. melanogaster* genome, encoding three distinct types of TPx—cytosolic, mitochondrial and secreted [65]. Five homologous genes for TPx have also been found in *A. gambiae* and *A. mellifera*; however, the gene for the secreted type of TPx appears to have been lost in *A. mellifera* in contrast to Diptera species [21]. The importance of TPx in stress protection has been shown in many studies dealing with different insect species. Stress conditions such as temperature changes, insecticide exposure or viral infection led to increased expression of TPx genes in *Bombyx mori* and *A. cerana*. In contrast, silencing of these genes led to faster death caused by fungal and viral infections in army worms (*Spodoptera* spp.) or pea aphids (*Acyrthosiphon pisum*) [64].

Methionine sulphoxide reductases (Msrs) catalyse the reduction of methionine sulphoxide to methionine and can thus reverse protein modifications caused by the oxidation of methionine groups [66,67]. In this process, protein methionine residues act as antioxidants because one ROS molecule is removed after each oxidation of a methionine residue by ROS and the reduction of methionine sulphoxide by the Msr enzyme. Through this mechanism, Msrs also participate in the redox regulation of protein activity [68].

## 3. Thioredoxin System of Insects

### 3.1. Thioredoxin System Is an Evolutionarily Conserved Component of Redox Regulation

The thioredoxin system has evolved in organisms across all evolutionary stages as the central disulphide reductase system that can provide electrons to a large spectrum of other proteins [16,17,69,70]. The thioredoxin system is essential for aerobic life and consists of thioredoxins (Trx), the enzyme thioredoxin reductase (TrxR) and the coenzyme NADPH. The Trx system represents an important cellular antioxidant mechanism, controls cellular redox balance and protects cells from oxidative damage. It also plays a vital role in many physiological processes. Components of the Trx system are found in the cytoplasm, chloroplasts, mitochondria and nucleus, and they may also be extracellular or membrane-bound [19,71].

The mechanism of the thioredoxin system reaction is based on the reactivity of Cys residues in the active site of the proteins. The thiol group (-SH) on the Cys side chain can be reversibly oxidatively modified, allowing the alteration of the protein structure and function. The thiol group serves as a nucleophile in the enzyme’s active site, as a binding site or to form covalent bonds to stabilise the protein [17]. Most protein thiols have pKa values between 8 and 9. They are thus protonated under physiological conditions, whereas thiols in the active sites of proteins more sensitive to oxidation are often characterised by lower pKa values, resulting in their deprotonation. The resulting thiolate anions are much more reactive than protonated thiols and readily form disulphide bonds with other proteins or glutathione. Disulphides are the most important oxidised form of thiols because they are stable and partially resistant to further oxidation [17,18].

The dithiol/disulphide reaction has become crucial for regulating metabolic processes in many organisms. In some proteins, a similar function is performed by selenocysteine, where the sulphur atom is replaced by selenium. The selenoate group in selenoenzymes is more reactive than cysteine thiolate and, therefore, more catalytically efficient [17]. TrxR is an example of an enzyme occurring in different Cys/Sec configurations: mammalian TrxR harbours Cys and Sec in its active site, whereas in the *Drosophila* TrxR Sec is replaced by another Cys residue [35].

TrxRs catalyse the transfer of electrons from NADPH to the oxidised form of Trx proteins, reducing the disulphide in the active site to the dithiol [72]. Reduced Trx can transfer reducing equivalents to diverse protein targets, e.g., the enzyme ribonucleotide reductase (RNR, EC 1.17.4.1) involved in DNA synthesis and repair, the antioxidant enzyme methionine sulphoxide reductase (Msr, EC 1.8.4.5) and peroxiredoxins (Prx, EC 1.11.1. 24) or enzymes involved in cell signalling pathways [17,73]. Specifically in insects, the reduction of oxidised glutathione by the Trx system is vital as insects lack GRs that perform this function in other organisms [35].

### 3.2. Insect Thioredoxins

Thioredoxins (Trx) are omnipresent small thiol proteins with a molecular mass of about 12 kDa. The Trx family includes cytosolic Trx (Trx1), mitochondrial Trx (Trx2) and a spermatid-specific Trx isoform (Sp-Trx3) [16]. The primary role of the Trx system is generally to provide a reducing environment and protect the cell from the deleterious effects of oxidative stress, which can eventually lead to apoptosis [73]. Trxs are also involved in DNA synthesis as electron donors for ribonucleotide reductase. Trx was first purified and described as a hydrogen donor for RNA in *E. coli* [74]. Numerous disulphide bonds involved in protein folding or the redox control of protein function are efficiently and selectively reduced by thioredoxins. In insects, the non-enzymatic reduction of GSSG to GSH by reduced Trx also occurs [13,14,72].

#### 3.2.1. Structure and Reaction Mechanism of Thioredoxins

Thioredoxins from different organisms contain a common basic structure called the thioredoxin fold, consisting of five β-sheets forming the protein core and four α-helices surrounding the central β-sheet [75]. The β-sheets and α-helices can be divided into N-terminal (β1, α1, β2, β3) and C-terminal (β4, α4, β5) connected by the α3 helix. The β-sheets at the N-terminus are oriented in parallel, and the β-sheets at the C-terminus in antiparallel [76]. The surface region at helix α3, where contact with TrxR occurs, exhibits a negative charge, a property common to eukaryotic thioredoxins. The active site with the typical Cys-Gly-Pro-Cys motif is localised on the protein surface between the β2 sheet and the α2 helix [69,77].

The thioredoxin fold and Cys-X-X-Cys motif are common in many thiol-dependent oxidoreductase proteins and enzymes, e.g., glutaredoxins and protein disulphide isomerases (PDI, EC 5.3.4.1), but also in functionally distinct enzymes such as glutathione peroxidases and glutathione transferases [78]. Cys residues are indispensable for the Trx redox functions; moreover, the substitution of Gly or Pro in the active site has a major effect on the stability and redox capacity of Trx. However, Gly or Pro substitution does not always result in a loss of activity, as demonstrated for mutated Trx from *E. coli* with Pro replaced by Ser, which shows higher activity than the native Trx form [76].

The reaction catalysed by thioredoxin can be described as a bimolecular nucleophilic substitution reaction, where Trx thiolate anion serves as the attacking nucleophile [79]. Electrons are transferred from Trx to the target protein, the disulphide bond of the target proteins is lost and a disulphide bond is formed in the oxidised form of Trx. Here, the thiol–disulphide conversion in Trx functions as a so-called redox switch between the oxidised and reduced form [18,80]. Thiol–disulphide exchange is influenced by several factors, such as the pKa of the redox-active Cys and the electrostatic environment of the surrounding amino acid residues [79]. Oxidised Trx are more stable than reduced Trx, with this difference in stability acting as a suitable driving force for the reducing reaction.

The Trx reaction starts with a nucleophilic attack of the N-terminal deprotonated thiolate on the disulphide of the target protein. The disulphide bond in the protein is broken, and an intermediary mixed disulphide is formed between Trx and the target protein. This first step of the reaction can proceed due to the low pKa of the N-terminal Cys compared to the pKa of the C-terminal Cys and free Cys residues in the surrounding environment. After mixed disulphide formation, the C-terminal thiol is activated by its deprotonation to thiolate. It attacks the neighbouring Cys at the Trx N-terminus, forming a disulphide bond in Trx and releasing the reduced target protein. The active thiolates acting during the reaction are stabilised by hydrogen bonds formed with surrounding amino acids. Finally, the oxidised Trx is converted back to its reduced state by the action of TrxR and NADPH [76].

#### 3.2.2. Specific Features of Insect Thioredoxins

Three basic types of Trxs and several thioredoxin-like proteins have been distinguished in model insect species. In *D. melanogaster*, three genes encoding Trxs (DmTrx-1, DmTrx 2 and DmTrxT) together with four other genes for thioredoxin-like proteins (Trx-like) have been identified. DmTrx-1 and DmTrxT are localised in the nucleus, whereas DmTrx-2 is found in the cytosol [21]. DmTrx-1 (also referred to as DHD, deadhead) is a Trx specific for the female ovary, while TrxT is a Trx specific for the male testis. The genes encoding both proteins are located as a gene pair on the X chromosome [81]. The cytosolic form DmTrx-2 is found in both sexes and is the most metabolically active, serving as a substrate for TrxR in its oxidised form. Bauer et al. [72] demonstrated that reduced DmTrx-2 was also the substrate in vivo for TPx, which points to a significant role for DmTrx-2 in maintaining redox balance in the cell. Conversely, glutathione reduction in *D. melanogaster* is mediated equally by DmTrx-1 and DmTrx-2. Under non-stress conditions, all three basic types of Trx are dispensable for life in *D. melanogaster*. Still, at elevated levels of ROS, DmTrx-2 has a positive effect in buffering oxidative stress [81].

The biochemical and physiological changes associated with the loss of DmTrx-2 were observed by Tsuda et al. [82], who found that mutant *D. melanogaster* individuals are more susceptible to the pro-oxidant paraquat; show higher expression of other enzymes, such as SOD or CAT; and contain more carbonylated proteins that are used as markers of oxidative stress. The deficiency of DmTrx-2 also caused a slight reduction in the lifespan of the animals, which corresponds with the observation that *D. melanogaster* mutants without the DmTrx-2 gene had a shorter lifespan than those without the mutation [81].

DmTrxs form dimers in oxidised and reduced forms, similar to human Trx [75]. The dimer formation of human Trx in vivo is promoted by oxidised and suppressed by reduced glutathione; thus, dimerisation and Trx functions can be regulated by the GSH/GSSG ratio. At high GSSG levels, indicative of oxidative stress, the function of the Trx system is reduced at the expense of the glutathione system. However, this is not the case for DmTrxs, which have a higher dissociation constant and in vivo occur predominantly as monomers. These DmTxrs properties are associated with the absence of GR since Trx is required for GSSG reduction in *D. melanogaster* along with the Trx system. If the *Drosophila* Trx system were suppressed in excess of GSSG, an even greater disbalance in the redox state and oxidative damage would occur.

Thioredoxins have also been investigated in other insect species. Three genes encoding one cytosolic (AgTrx-1) and two mitochondrial (AgTrx-2 and AgTrx-3) forms of Trx were found in the genome of the mosquito *A. gambiae*. The honey bee genome also contains three Trx genes: mitochondrial AmTrx-1 and cytosolic AmTrx-2 and AmTrx-3 [21]. Yao et al. [83] investigated the expression of the AccTrx-2 gene in the eastern honey bee (*Apis cerana cerana*) at different tissues and developmental stages or under stress conditions. The highest expression of AccTrx-2 was found in brain tissue, consistent with the fact that brain cells are more sensitive to oxidative damage than other tissues. An analysis of individual developmental stages showed an increased expression in 4-day-old larvae and in adults 15 days after hatching. Unlike pupae, larvae and especially adults have higher metabolic activity and ROS production. Abiotic factors such as temperature changes or treatment with H_2_O_2_, herbicides and insecticides increased the gene expression of AccTrx-2.

Analogous to DmTrx, two Trxs, the cytosolic SlTrx-1 and the mitochondrial SlTrx-2, were identified in the tobacco cutworm (*Spodoptera litura*), whereas only one Trx (BmTrx) was found in the silkworm (*Bombyx mori*). The gene expression for both SlTrx and BmTrx increased after H_2_O_2_ injection and in response to stress factors such as temperature changes or infection by microorganisms, confirming the antioxidant and protective role of these proteins [84,85]. In the cotton bollworm (*Helicoverpa armigera*), gene expression of Trx-1 was increased after exposure to stress factors such as high and low temperature, UV radiation, mechanical injury or parasite infestation [86]. Interestingly, Trx from the Indian meal borer (*Plodia interpunctella*) is classified as a type I allergen and acts like the human allergen thioredoxins from wheat and maize [87].

In some organisms, including insects, Trx-like proteins have been identified that may contain one or more domains in common with typical domains in the Trx structure. Trx-like proteins have an N-terminal Trx domain but a C-terminal domain non-homologous to other proteins and with unknown function, and only some of them contain the conserved Cys-Gly-Pro-Cys motif of the Trx active site. One of the human Trx-like proteins, Trx-like-1, serves as a substrate for TrxR-1, and its homologs have also been found in *A. mellifera, D. melanogaster* and *A. gambiae* [21,88]. Lu et al. [89] characterised a gene encoding a Trx-like protein (AccTrx-like1) in *A. cerana cerana* and monitored its expression. They found that among bees at different developmental stages, the expression of AccTrx-like1 was highest in larvae and relatively low in adults. The larvae probably have not yet developed more complex antioxidant systems and use Trx-like proteins for their protection. Although the overall expression of AccTrx-like1 was low in adults, tissue-specific increased expression was observed in; sites with higher ROS abundance and a need for active antioxidant mechanisms such as the epidermis and brain tissue In a follow-up experiment, AccTrx-like1 expression was found to be induced by low temperature and short-term exposure to H_2_O_2_, suggesting protective function against cold-induced stress and a rapid response to short-term oxidative stress [89].

### 3.3. Insect Thioredoxin Reductase

#### 3.3.1. Structure of the Active Site and the Catalytic Mechanism of TrxR

Thioredoxin reductases are homodimeric flavoenzymes catalysing electron transfer between NADPH and redox-active disulphides. The same family of oxidoreductases also includes glutathione reductases and lipoamide dehydrogenase (EC 1.8.1.4). Evolutionarily, two types of TrxRs have evolved: archaea, bacteria, plants and fungi have TrxRs with low Mr subunits (35 kDa), whereas TrxRs in higher eukaryotes are composed of high Mr subunits (55 kDa) [17,34]. High Mr TrxRs evolved from GRs already in lower eukaryotes and coexisted with low Mr TrxRs [90]. During further evolution, high Mr TrxRs were conserved only in animals but disappeared in plants and fungi, where only low Mr TrxRs have been conserved. A comparison of the two types also shows that high Mr TrxRs show higher homology to other enzymes in the family (e.g., GR) than to low Mr TrxRs. At the same time, high Mr TrxRs are more diverse within their family than low Mr TrxRs, which tend to be more conserved [69,91].

Significant differences exist in the structure and mechanism of transfer of reducing equivalents between low- and high-molecular-weight TrxRs. During the catalytic mechanisms of low Mr TrxRs, the conformation of one of the enzyme domains changes. In the first conformation, the redox-active disulphide is reduced by a flavin cofactor, followed by rotation, transferring the resulting dithiol to the enzyme’s surface where it can reduce Trx. Simultaneously, when the dithiol is located on the enzyme surface, NADPH is in a suitable position to reduce the oxidised FAD cofactor.

Each high Mr TrxR monomer contains three domains: a FAD-binding domain, a NADPH-binding domain and a domain that provides an interface between the two monomers. Active sites consisting of FADs are located on both monomers, N-terminal redox disulphide and C-terminal redox disulphide. TrxRs with low Mr lack the C-terminal disulphide and thus have only two active sites [92]. The active site of the high Mr TrxR from *D. melanogaster* contains three typical redox centres: FAD, an N-terminal disulphide adjacent to flavin (Cys57-Cys62) and a C-terminal disulphide (Cys489′-Cys490′), which is, however, located on a different subunit than the first two mentioned centres. Thus, the active site of Trx dimer is composed of components from both polypeptide chains.

The polypeptide chain containing the C-terminal disulphide is flexible. It first occupies a position near His464′, which serves as a basic catalyst, to form the mixed disulphide Cys57-Cys490′ and then reaches the protein surface for reaction with the substrate. The mutant variant of DmTrxR lacking His464′ showed activity decreased to 2% of the value for the wild-type enzyme; therefore, His464′ appears to be an essential acid-base catalyst for the dithiol–disulphide reaction [93]. Glutamate residues also play an auxiliary role in catalysis, specifically Glu469′ and Glu470′ in DmTrxR. Glu469′ is more important for catalysis and likely increases the basicity of His464′ by facilitating its orientation to Cys57 [93].

In contrast to low Mr Trxs, high Mr TrxRs transfer reducing equivalents from the first disulphide in the interior of the enzyme to the second disulphide or selenosulphide on the enzyme surface in a way that no significant conformational change is required [34]. The transfer of reducing equivalents during catalysis proceeds from NADPH to FAD, then via the N-terminal disulphide to the C-terminal disulphide of the second subunit and finally to Trx [92,94]. The C-terminal disulphide is a crucial component for the functioning of the high Mr TrxR. Still, this end motif has different forms in different organisms, e.g., Ser-Cys-Cys-Ser in *D. melanogaster*, Thr-Cys-Cys-Ser in *A. gambiae*, Gly-Cys-Cys-Gly in nematodes and Gly-Cys-Sec-Gly in mammals [95,96]. The occurrence of selenocysteine instead of cysteine in mammalian enzymes has been considered an evolutionary development because the chemical properties of selenium generally enhance the catalytic activity of oxidoreductases. However, it has been suggested that TrxR containing Sec evolved evolutionarily earlier than Cys-containing TrxR [90]. The presence of selenium in the form of selenoate in the active site may accelerate the reaction, with selenoate being a stronger nucleophile than thiolate in attacking the disulphide bond of Trx and a better electrophile than disulphide in accepting electrons from the N-terminal Cys [97]. Alternatively, the evolutionary pressure to maintain the complex selenocysteine insertion machinery might be related to the increased resistance of selenoenzymes to inactivation by irreversible oxidation [98,99]. This resistance could be provided by the higher ability of the oxidised form of selenocysteine, seleninic acid, to be reduced back to Sec compared to the reduction of the cysteine sulphinic group to a thiol.

In the case of insect TrxR, which does not contain Sec, it is therefore necessary to compensate for the lower reactivity of Cys. The increase in the nucleophilicity of Cys490′ attacking the substrate in DmTrxR is achieved by stabilising the resulting thiolate with serine residues flanking the reactive Cys [100]. In contrast, the increase in the electrophilicity of Cys490′, which forms a disulphide with Cys489′, is accomplished by the polarisation of the disulphide bond by the positive charge of His464′. Thus, Cys490′ readily accepts electrons from Cys at the N-terminus and equals Sec in its catalytic properties. Thus, the reactivity of DmTrxR is significantly influenced by the structure and geometric arrangement of amino acids in the active site [101]. The ionisation state of the functional groups of amino acids is strongly affected by the pH of the environment. Free thiol groups have a pKa of around 8.5, whereas selenoate has a pKa of around 5.3. Sec ionises at physiological pH, but Cys is still in a protonated state, and the influence of neighbouring amino acid functional groups is needed for its deprotonation. Within the pH range of 6–9, TrxR with a Cys-Sec motif showed minimal fluctuations in the enzymatic activity. In contrast, TrxR containing two Cys was observed to strongly decrease activity at pH lower than 7 [100]. To further uncover the function of Sec in the catalytic mechanism of TrxR, several attempts have been made to substitute Cys and Sec in the active site. The substitution of Cys for Sec in DmTrxR did not cause a significant change in activity when Trx was reduced, demonstrating that the presence of a Sec residue is not required to catalyse the reduction of the disulphide bond in Trx. A possible explanation is the proximity of Ser in the C-terminal sequence of DmTrxR, which contributes to disulphide reactivity [102]. In contrast, the substitution of Sec for Cys in mammalian TrxR led to a 100-fold reduction in enzyme activity [100,103]. However, when using a different substrate, e.g., selenocysteine or selenite, the activity of mammalian TrxR with a Cys-Cys motif was only 2–4-fold lower compared to native DmTrxR [104]. The activity of the enzyme can also be affected by disruption of the C-terminal motif with active Cys or Sec, which, in the oxidised form, forms an octahedral ring. After the insertion of Ala between the Cys-Se pair, the activity of the mammalian enzyme was reduced 4–6-fold, whereas DmTrxR containing the Cys-Cys pair had 100–300-fold lower activity after Ala insertion. Thus, the active centre of DmTrxR appears to be more dependent on the structure of the eight-membered ring than mammalian TrxR.

TrxRs are, in general, enzymes with broad substrate specificities. Although this extended substrate spectrum has been attributed mainly to the presence of Se in the active site of the enzyme, some low-molecular-weight compounds have been shown that the C-terminal reducing centre cannot reduce them but instead can react directly with the N-terminal centre. The reduction of these substances is thus independent of the Se present in the enzyme’s active site [104]. In addition to Trx, mammalian TrxR can directly reduce other macromolecular substrates such as PDI, glutaredoxin, GPx or granulysin. The TrxR substrates also include small molecules such as H_2_O_2_, lipid hydroperoxides, ubiquinone and other quinones, dehydroascorbate, lipoic acid and lipoamide, S-nitrosoglutathione, selenodiglutathione, selenite, selenocystin and 5,5′-dithiobis-2-nitrobenzoic acid (DTNB). DTNB, ubiquinone, lipoic acid and lipoamide are directly reduced by the reaction centre at the N-terminus. Se-containing substrates are also reduced by a mutant mammalian enzyme with a Cys-Cys motif at the C-terminus or by the N-terminal reaction centre itself without significant loss of activity. In insects lacking GR, the glutathione reduction is provided by the Trx system when GSSG is reduced by Trx, which is subsequently restored by the action of TrxR and NADPH. TrxR cannot directly reduce GSSG despite GSSG having a more positive redox potential. The reduction most likely cannot proceed due to the poor availability of the C-terminal active site for the GSSG molecule [105].

#### 3.3.2. Specific Features of Insect Thioredoxin Reductases

Although the key enzymes of the antioxidant system are conserved in animals through evolution, special variations of this system have evolved in certain groups. The main difference between insects and other animals is the loss of genes for GR and GPx, which are important antioxidant enzymes [21]. Their functions are represented by TrxR and TPx [35,106]. TrxRs are, therefore, an important subject of investigation in insects. The *D. melanogaster* genome contains two genes for TrxR. TrxR-1 encodes three TrxR isoforms, two cytosolic and one mitochondrial, which differ in N-terminal chain structure in addition to localisation. TrxR-2 has not yet been thoroughly studied; the mitochondrial localisation of the encoded protein is assumed. A mutation of either gene is lethal to the organism at the larval developmental stage or shortly after reaching adulthood [107]. In *A. gambiae*, only one TrxR gene has been found encoding three variants homologous to those in *D. melanogaster*, with AgTrxR 1 showing 69% sequence identity to DmTrx-1 [96]. Only one TrxR gene has been identified in *A. mellifera* where the mitochondrial form was not confirmed [21].

Bauer et al. [92] described the catalytic cycle of DmTrxR-1 in detail. The enzyme in the oxidised state contains a disulphide adjacent to a flavin (Cys57-Cys62) near the disulphide at the C-terminus of the polypeptide chain (Cys489′-Cys490′). On activation, the first NADPH molecule forms a Michaelis complex with the enzyme, which subsequently gives rise to an equilibrium mixture of four forms reduced by two electrons. Charge transfer between the C- and N-terminal disulphides is accomplished by Cys57 and Cys490′, which form a mixed disulphide. A reaction with a second NADPH molecule produces a second Michaelis complex, and then the enzyme is reduced to a four-electron state. These steps are called reductive reactions in the cycle, whereas the following are called oxidative reactions. After dissociation of oxidised NADP+, oxidised Trx reacts with the reduced enzyme to form a mixed disulphide, which is also the first step for dithiol–disulphide exchange, at the end of which reduced Trx is released, and DmTrxR returns to the initial state and can initiate the next catalytic cycle [105].

In the aquatic insect harlequin fly (*Chironomus riparius*), TrxR 1 gene expression was induced after exposure to paraquat and cadmium chloride. Here, cadmium was used as a representative xenobiotic for aquatic animals in the environment, which is known to induce oxidative stress in exposed cells [108]. Krishnan et al. [109] confirmed the presence of both TrxR-1 and TrxR-2 in the gut of larvae and adults of the Colorado potato beetle (*Leptinotarsa decemlineata*). The relative gene expression of TrxR-1 was higher in larvae, whereas TrxR-2 expression was without significant differences. In *H. armigera*, the expression of TrxR-1 was increased after exposure to various stress factors such as high and low temperature, UV, mechanical wounding and parasite infestation [86].

#### 3.3.3. Methods for Determination of TrxR Activity in Insects

Although TrxRs have been known since the last century, their activity in insects has not yet been investigated in detail. Most of the published methods to determine the activities of these enzymes have been performed in bacteria, plants or mammals. However, bacterial and plant TrxRs fundamentally differ from mammalian TrxRs as they are low Mr enzymes that do not contain the C-terminal disulphide and Sec in the enzyme’s active site [110]. Mammalian TrxRs also reduce a broader range of substrates than bacterial and plant TrxRs [111]. Both mammalian and insect TrxRs are classified as having higher Mr; however, a different catalytic reaction mechanism is predicted for insect TrxRs due to the absence of Sec.

In addition to its natural substrate Trx, TrxR is able to reduce a diverse range of other substrates, which has allowed the development of various methods for determining the activity of this enzyme [112]. The efficiency of each method depends on the type of TrxR and Trx or other substrate. TrxRs from different organisms have different affinities for the target molecules and for Trx itself, so it is best to perform the analysis with TrxR and Trx from the same organism [111].

Several spectrophotometric methods with different substrates have been established for the determination of mammalian TrxR activity. One of the most commonly used substrates is Ellman’s reagent (5,5′-dithiobis-2-nitrobenzoic acid, DTNB). In the TrxR-catalysed reaction, DTNB is reduced by NADPH to the yellow product 5-thionitrobenzoic acid (TNB), which has an absorption maximum at 412 nm. The unit of TrxR activity is then defined as the amount catalysing the reduction of 1 μmol of DTNB (i.e., the formation of 2 μmol of TNB) per minute. The DTNB method is advantageous due to its rapidity, sensitivity due to a high extinction coefficient of the TNB, product simplicity and low cost [112]. However, other thiols present in cell extracts can also reduce DTNB; therefore, purified enzymes are preferred over cell lysates [111]. If the analysed sample also contains GR and GSH, these must first be removed from the sample; otherwise, GR and GSH would also reduce DTNB and increase the measured values of TrxR activity [113]. Using specific mammalian TrxR inhibitors, most commonly gold-containing compounds such as aurothioglucose or auranofin, the background of TrxR-independent DTNB reduction can be easily measured [112]. The reactions with DTNB can also be used for the detection of TrxR activity on polyacrylamide gels after separation by native electrophoresis, and the intensity of formed bands can be strongly enhanced by other dyes [114].

Insulin is another frequently used TrxR substrate, and the spectrophotometric insulin method is considered the most reliable assay for TrxR activity [111]. Trx reduced by NADPH-dependent TrxR activity can subsequently reduce two disulphides of insulin, and then the consumption of NADPH can be measured by a decrease in absorbance at 340 nm. Determining TrxR activity from cell lysates is also possible by combination with the DTNB method. After insulin reduction, the reaction is stopped with guanidine hydrochloride and the thiol groups of insulin are determined by DTNB. Once a calibration curve has been generated using TrxR standards, the enzyme activity in a given sample can be determined. This combined method is more specific than using DTNB alone. Alternatively, the TrxR activity with insulin as the substrate can be measured by monitoring the NADPH consumed at 320 nm, but here, interference with NADPH-metabolising enzymes can occur in biological samples [113]. Using the same principle of following NADPH consumption, mammalian TrxR activity can be measured with a wide range of substrates, e.g., L-cystine, L-selenocystine, selenite, juglon and others [111,115,116]. The highest reaction specificity can be obtained when using purified Trx of a given organism as a substrate, but this procedure is costly and not always feasible. As an alternative, commercially available Trx from *E. coli* or the parasitic protozoan *Plasmodium falciparum* have been used to measure human TrxR activity [112].

The ability of mammalian TrxR to reduce selenocystine, a diselenide-bridged amino acid, was exploited to develop a continuous kinetic assay compatible with a 96-well plate-based format [117]. The TrxR is assessed by the consumption of NADPH at 340 nm, and the authors reported among its advantages observed compatibility with nonionic detergents such as NP-40, commonly used in lysis buffers and known to inhibit TrxR activity in the insulin end-point assay. Montano et al. [118] developed a highly sensitive fluorescence assay to determine human TrxR activity based on the reduction of fluorescein isothiocyanate (FITC)-labelled insulin, which shows a higher affinity for Trx than unmodified insulin. The increased fluorescence emission at 520 nm is measured upon the binding sites’ reduction in the labelled insulin. Compared to the conventional insulin assay, the method using FITC-labelled insulin is more sensitive and more suitable for determining TrxR in biological samples. The method has been tested in human plasma samples, serum, cancer cells and isolated lymphocytes.

Small 14 kDa thioredoxin-related protein TRP14 has been identified as an interesting TrxR substrate. TRP14 belongs to the thioredoxin family and is reduced by TrxR, but unlike Trx, it does not interact with most of its natural substrates, including insulin or peroxiredoxins. However, TRP14 can reduce L-cystine several times more efficiently than Trx. Therefore, a method has been developed using fluorescently labelled L-cystine, where the fluorescence released by the reduction corresponds to the consumption of TRP14 and thus TrxR activity [119]. However, similar to other mentioned substrates, measurement with TRP14 is not recommended in cell lysates, as other enzymes, including GR, can also reduce this protein substrate [111].

TrxR inhibitors are used with the subsequent subtraction of absorbance changes in the presence of the inhibitor for more accurate results in spectrophotometric measurements. A high affinity for noble metals characterises the active Cys and Sec TrxRs; therefore, gold- or platinum-containing compounds can be used as inhibitors [120]. As substances containing gold atoms, Auranofin and aurothioglucose represent the most specific and commonly used irreversible inhibitors of animal TrxRs acting in vitro at nanomolar concentrations [91,121]. On the other hand, auranofin appears to be only a weak inhibitor of DmTrxR [35]. Organometallic gold compounds with oxidation number I or III and cis- and trans-platinum derivatives have also proven to be effective inhibitors of animal TrxR. Compounds based on mercury, ruthenium, cadmium, copper and zinc showed a less efficient ability to inhibit TrxR compared to gold- and platinum-containing inhibitors [120]. Some dinitrohalobenzenes, organochalcogens, naphthalazine derivatives, thiol alkylating agents, arsenic compounds, dicarboxylic acids, polyphenols, quinones, nitroaromatic compounds, nitrosoureas and curcumin can also inhibit TrxR to varying degrees [91,120,121]. Trx inhibitors are highly sought after, finding use mainly for therapeutic purposes in cancer treatment. The tested Trx inhibitors are imidazolyl disulphide PX-12, cyclohexadienone AW 464 or pleurotin derivatives [70].

Monitoring mammalian TrxR activity in situ in living cells has been achieved by developing various fluorescent probes. Fluorescent probes are generally characterised by high sensitivity and specificity and non-invasive and rapid analysis when investigating cellular processes; however, the design and synthesis of high-quality probes are more complex [122]. The first specific probe to detect intracellular TrxR activity was a fluorogenic substrate TRFS-green, which contains naphthalimide as a fluorophore bound to an artificial cyclic disulphide 1,2-dithiolane. Upon the reduction of TRFS-green by TrxR, a green fluorescent signal is emitted [123]. Later, other probes based on TRSF, such as TRSF-red, with emission wavelengths shifted above 600 nm, and the two-photon probe TP-TRFS, were reported. However, ambiguity in the specificity and efficiency of these probes in vivo has been reported, and their wider use for TrxR studies requires further testing and validation [122].

### 3.4. Regulation of Thioredoxin System

The activity of the Trx system can be regulated in several ways, including gene expression, post-translational modifications and protein–protein interactions. These regulatory mechanisms allow the system to respond to changes in the cellular redox environment and adapt to changing conditions. Various transcription factors, such as nuclear transcription factor 2 (Nrf2), TATA-binding protein or cAMP response-binding protein (CREB protein), are involved in the regulation of mammalian Trx and TrxR gene expression. These transcription factors are activated by stress factors such as oxidative stress or inflammation. Both Trx and TrxR genes contain enhancer sequences called antioxidant response elements (AREs) in the promoter region, to which Nrf2 binds under oxidative stress conditions. AREs are also present in genes encoding other proteins involved in antioxidant protection [124]. Se availability may also affect the expression of genes for mammalian TrxR containing Sec. An increased supply of non-toxic doses of Se induces expression, whereas Se deficiency decreases it. Toxic levels of Se act as pro-oxidants and inhibit TrxR activity. Other substances that increase TrxR expression include estrogens or Nrf2 activators such as acrolein, peroxynitrite and cadmium [73].

Post-translational modifications also regulate the activity of the Trx system. As mentioned above, Trx can undergo S-nitrosylation, i.e., the reaction of NO with the thiol group of Trx reducing its reactivity [125,126,127,128]. The same negative effect on Trx activity is exerted by glutathionylation, where a mixed disulphide of glutathione and another protein, Trx, is formed. Both nitrosylation and glutathionylation are reversible processes [129,130,131]. Evidence for post-translational modifications in TrxR has not yet been reported. Both Trx and TrxR can be modified by acetylation, more precisely by binding an acetyl group to Lys. The acetylation of human TrxR-1 has been shown to prevent the oligomerisation and oxidation of the enzyme associated with a loss of activity [132,133].

The Txnip protein negatively regulates Trx under stress conditions. Txnip expression is increased during oxidative stress, when Txnip binds to thiols in the active site of reduced but not oxidised Trx, forming an inactive mixed disulphide. Txnip also functions in a TRX-dependent manner as a positive regulator of apoptotic cell death by activating the apoptosis kinase cascade [134].

### 3.5. Enzymes Dependent on the Thioredoxin System

The main role of Trx and the Trx system is to provide reducing equivalents to other proteins. These enzymes are regenerated from their oxidised state by the Trx system and are directly dependent on its reducing capacity; thus, their activity can be regulated by the Trx system. The most important Trx-dependent enzymes include ribonucleotide reductases (EC 1.17.4.1), peroxiredoxins (EC 1.11.1.24) and methionine sulphoxide reductases (EC 1.8.4.5).

#### 3.5.1. Ribonucleotide Reductase (RNR)

RNR enzyme in *E. coli* was the first protein discovered to accept electrons from Trx. RNR is an important enzyme that catalyses the conversion of ribonucleotides to deoxyribonucleotides, which are the building blocks for DNA synthesis and repair [74,135,136]. RNRs can be divided into several classes based on the type of cofactor used and the mode of radical formation during the catalytic mechanism. All eukaryotes, from yeast to humans, contain class I RNR, whereas class III includes enzymes from anaerobic organisms that can be inactivated by oxygen [137]. The RNR mentioned above from *E. coli* is classified as a Class I enzyme and requires oxygen to form a stable tyrosyl radical via the Fe-O-Fe centre. Class II RNRs do not need or are inhibited by oxygen and generate the thiyl radical by adenosylcobalamin. During catalysis, in addition to the radical, a disulphide is formed from two cysteine thiols in the active site of the enzyme, which must then be reduced to its original state. Thus, RNRs depend on external thiol reductases required to restore the enzyme to its initial active configuration and continue the next cycle of catalysis. The electron donors for class I RNRs are Trxs or glutaredoxins, which appear to be more efficient donors, as confirmed in *E. coli* or mice [135]. In most organisms, oxidised glutaredoxin is reduced by glutathione, which is subsequently reduced by glutathione reductase, whereas insect GSSG is reduced to GSH by the Trx system [21]. GSH levels may then directly influence RNR activity and cell proliferation rates through the availability of deoxyribonucleotides. This link between GSH regeneration and RNR activity supports the suggestion that reduced GSH levels appear to be a possible cause of the accumulation of damaged DNA [135].

#### 3.5.2. Peroxiredoxins

The peroxiredoxin family includes proteins found in prokaryotic and eukaryotic organisms, which reduce H_2_O_2_ and organic peroxides using Cys thiol groups in the active site. Due to their active role in removing H_2_O_2_, organic peroxides and, in some cases, peroxynitrite, Prxs are recognised as key antioxidant enzymes protecting organisms from oxidative stress [138]. Beyond direct ROS scavenging, Prx functions include also interfacing with other components of the antioxidant systems and maintaining redox balance. Prxs use Trx or GSH for the regeneration of reduced thiols; for this reason, these enzymes are also referred to as thioredoxin peroxidases (TPx) and glutathione peroxidases (GPx), respectively. In insects lacking GPx, it is assumed that TPx can use both Trx and GSH as substrates [106]. In many studies, the term peroxiredoxin is used as a synonym for TPx because of the history of the discovery of this enzyme. In higher eukaryotes, Prxs are classified into six groups, which can be further divided into three subgroups based on the number of Cys residues in the active site of the enzyme into 1 Cys, typical 2 Cys and atypical 2 Cys Prxs. Typical 2 Cys Prxs contain two conserved Cys residues at the C-terminus and N-terminus. In contrast, atypical 2 Cys Prxs have only one conserved Cys at the N-terminus and group 1 Cys Prxs proteins contain only N-terminal Cys [138]. Oxidised Prxs are reduced by Trx, which is subsequently reduced within the Trx system. As the reduced Trx is also involved in reducing GSSG and other molecules, changes in Prx levels also affect the Trx system due to their interdependence [139].

Five genes encoding Prx have been found in *D. melanogaster* [106]; moreover, a sixth putative Prx homologous gene was later reported by Corona et al. [21] in honey bees. The mitochondria of *D. melanogaster* contain peroxiredoxin 3 (DmPrx3) and peroxiredoxin 5 (DmPrx5). DmPrx3 is localised exclusively in the mitochondrion, whereas DmPrx5 is also found in other compartments. These are the only enzymes capable of reducing H_2_O_2_ and organic peroxides in the internal environment of the fly mitochondria, as they do not contain GPx fulfilling this role in mammalian mitochondria. For these reasons, various consequences of DmPrx removal have been investigated. The reduced expression of either DmPrx5 or DmPrx3 by 90–95% caused no or only a slight change in the life span of *Drosophila* flies and their sensitivity to oxidative stress. However, when the expression of both enzymes was reduced simultaneously, the life span was shortened by 80% and thiol homeostasis was severely disrupted due to an increase in oxidised protein thiols and GSSG levels, followed by induced apoptosis in fly muscles and digestive tissues [139].

Five genes encoding Prx/TPx homologous to DmPrx were identified in *A. mellifera*, with the exception of extracellular TPx-2. AmTPx-3 is found in mitochondria, and types 1, 4 and 5 are localised in the cytosol [21]. Various studies have investigated the expression of TPx genes in the eastern honey bee (*A. cerana cerana*). The expression of AccTPx-1, AccTPx-3, AccTPx-4 and AccTPx-5 was increased by abiotic stressors such as low and high temperature, exposure to H_2_O_2_, UV or other oxidants and insecticides, confirming the antioxidant role of these enzymes in the honey bee. However, the expression levels varied among bee body parts and developmental stages, so it was impossible to generally pinpoint the site of the highest expression of all TPx, and each TPx isoform should be considered separately [140,141,142,143].

In addition to antioxidant function, other roles of Prx have been identified in the regulation of redox status and signalling. Prx serves to regulate the level of H_2_O_2_ involved in intracellular signalling, which propagates through oxidative modifications of protein thiols [6,14,15]. Prxs might be specifically and temporarily inactivated to allow local accumulation of H_2_O_2_ to fulfil their signalling role. A reversible loss of peroxidative activity can be achieved by hyperoxidation, where the sulphenic group (Cys-SOH) of peroxiredoxin is further oxidised to a sulphinic group (Cys-SO_2_H) instead of forming a disulphide or by phosphorylation of specific Tyr or Thr residues [138]. However, this opens a question of how H_2_O_2_ might react with the target protein thiols in the intracellular environment with other thiols that compete with these proteins. Thus, it has been suggested that H_2_O_2_ may not react directly with the signalling proteins but instead use Prx as a mediator of signal transduction. Mammalian Prxs have been confirmed to have an active role in protein oxidation and to directly catalyse the formation of disulphide bonds, specifically in the case of ASK1 activation and the MAPK pathway leading to apoptosis. Evidence supporting a direct catalysis instead of a hyperoxidation mechanism is provided by identifying a disulphide exchange intermediate between Prx-1 and ASK1. Surprisingly, in contrast, mammalian Prx-2 inhibits the activation of this pathway and thus acts antagonistically to Prx-1 [144].

#### 3.5.3. Methionine Sulphoxide Reductase (Msr)

Msrs are ubiquitous enzymes that catalyse the conversion of free and protein-bound methionine sulphoxide (MetO) to methionine (Met), thus playing a role in protein regulation and antioxidant protection. The oxidation of Met residues in proteins can alter protein properties and function, and at the same time, this oxidation can also affect protein phosphorylation. It has been shown that when accessible Met residues adjacent to phosphorylation sites on an enzyme are oxidised, the phosphorylation of these sites can be inhibited [66,67].

Trx subsequently regenerates the enzyme because it contains Cys residues in the active site that are involved in the catalytic mechanism. ROS can oxidise Met to form MetO in a similar manner to how Cys is oxidised and can, therefore, be used for ROS scavenging in antioxidant defence and for cellular regulation through redox reactions [145,146,147]. Two main types of Msrs, MsrA and MsrB, are found in multicellular organisms with several different isoforms. Both types are stereospecific, MsrA reduces methionine-S-sulphoxide, and MsrB reduces methionine-R sulphoxide [67]. Single genes for both MsrA and MsrB were identified in the genomes of *D. melanogaster*, *A. mellifera* and *A. gambiae* [21].

Theories of ageing and reduced lifespan are also related to oxidative stress. Although it has been suggested that the increased expression of the MsrA gene in *D. melanogaster* or the knockout of the MsrA gene in the mouse increases the lifespan of these animals, more recent studies show different conclusions [66,148]. Bruce et al. [149] characterised the in vivo effects of Msr deficiency in *D. melanogaster*. Individuals lacking either MsrA or MsrB did not display any significant phenotype, and shortened lifespan and slower larval development were observed only when both Msr variants were deficient. Salmon et al. [150] showed that although mice lacking MsrA were more susceptible to oxidative stress damage, the lifespan of the mice tested was not affected. As different opinions and studies are conflicting, the effect of Mrs on life extension in some animals is still not fully elucidated.

## 4. Functions of the Trx System in Insects under Physiological and Stress Conditions

The Trx system is an important disulphide reductase system in insects involved in the redox control of various signalling pathways through interactions with a large number of proteins. Cys residues of proteins sensitive to oxidative modifications participate in various cellular signalling pathways through interactions with other proteins. Any significant changes in the activity of Trx or TrxR will also affect the activity of redox-controlled proteins dependent on the Trx system to regenerate their redox capabilities, ultimately leading to increased amounts of oxidised proteins. Reducing the capacity of Trx is dependent on the activity of TrxR and the availability of NADPH. This is a hierarchical arrangement of redox-active compounds ensuring the proper functioning of the system and proper redox control of the cells [73]. However, the target proteins of the Trx system can also be regulated by other mechanisms, such as substrate binding, allosteric effectors or post-translational modifications, leading to changes in metabolic pathways [17,126].

The antioxidant effect of Trx manifests itself in two crucial ways. First, Trx serves as an electron donor for peroxidases that scavenge ROS, thereby reducing the extent of lipid peroxidation, DNA damage and protein dysfunction under increased ROS levels. Second, Trx acts as a disulphide reductase and reduces the oxidised disulphide bonds of other proteins (e.g., kinases, phosphatases, and transcription factors) to restore their physiological function (Figure 2). The involvement of Trx in the defence against oxidative stress and the maintenance of redox homeostasis has been demonstrated in *B. mori*, *A. mellifera* and *A. cerana cerana*. The increased expression of genes for TrxR in *A. mellifera* and *A. cerana* was confirmed to be induced by temperature stress and the oxidants H_2_O_2_ and methyl viologen [151], while in *A. cerana cerana* it was due to temperature change and UV irradiation [152]. TPx gene expression was also significantly increased in *B. mori* larvae after exposure to different stress conditions such as high and low temperature, H_2_O_2_ administration and viral infection [153]. In the aquatic species *Chironomus riparius*, CrTrxR gene expression was increased after exposure to cadmium and paraquat, which can be considered as a biomarker of oxidative stress induced by environmental contaminants [108]. In addition to oxidative stress, Trx in mammalian cells is also involved in regulating nitrosative stress caused by reactive nitrogen species, where nitric oxide binds to the thiol residues of proteins in a process called S-nitrosylation [73,128].

### 4.1. Redox Balance and Antioxidant Defence

The antioxidant activity of the Trx system is mainly based on electron transfer to peroxiredoxins, methionine sulphoxide reductases and specific redox-sensitive transcription factors [69]. Prx serves to reduce H_2_O_2_ and organic peroxides, while Msr serves to reduce methionine sulphoxide to methionine. Both enzymes require Trx to regenerate thiol groups in the active site to allow the next catalysis cycle to proceed [139,145]. Prxs have a high catalytic efficiency in reducing H_2_O_2_ and a reaction rate of 10^7^ M^−1^ s^−1^, which is several times higher than other thiol proteins outside the peroxiredoxin family [154]. The antioxidant function of Prx has been investigated and confirmed in both *D. melanogaster* and *A. cerana* [106,140]. TrxR can directly reduce H_2_O_2_ and lipid peroxides, but this reaction occurs only as an alternative pathway at high peroxide levels [155]. Free protein methionines can also be oxidised due to oxidative stress, which can affect the function of oxidised proteins. The methionine sulphoxide formed by oxidation is reversibly reduced by the action of Msrs, which are thus involved in the antioxidant protection of the cell [145]. The Trx system is also an electron donor for the enzyme RNR, which is required for the formation of deoxyribonucleotides. Thus, the Trx system may contribute to the repair of DNA damaged by ROS [69].

Glutathione is considered the main thiol–disulphide redox buffer of the cell. The usual concentrations of GSH in the cytosol are within the range of 1–10 mM, which is significantly higher than most other redox-active compounds; therefore, the levels of reduced (GSH) and oxidised forms (GSSG) of glutathione is used to estimate the redox state of the cellular environment. Another important thiol system in the cell is thioredoxin, whose concentration in the cell is in the units of μM [156]. Both of these systems are linked to NADPH, an essential electron donor for redox metabolism, and to TrxR, which in insects regulates the levels of both GSH and Trx and the redox balance in the cell [35,156].

The antioxidant functions of ascorbate are based mainly on ROS scavenging and the regeneration of the reduced form of tocopherols, which are involved in reducing lipid peroxidation. The oxidation of ascorbic acid by superoxide, hydroxyl or other radicals produces monodehydroascorbate, which can be further oxidised to dehydroascorbate or reduced back to ascorbic acid. In animals, the NAD(P)H-dependent reduction of monodehydroascorbate is achieved directly by cytochrome b5-reductase (EC 1.6.2.2) or TrxR [157]. The reduction of dehydroascorbate back to ascorbic acid is mediated by GSH, an electron donor, catalysed by the enzyme dehydroascorbate reductase (EC 1.8.5.1). However, in organisms lacking GR, TrxR is also important for this reaction to restore the reduced GSH level. Another possible link to components of the antioxidant system may be the ability of TrxR to reduce lipoic acid and lipoamide, which can scavenge some ROS [158].

### 4.2. Ageing and Life Span of Insects

According to the theory of oxidative ageing, the major contribution to ageing and possible death of the organism is caused by increasing damage to cellular macromolecules and their accumulation due to the negative effects of ROS [22,159]. However, it has gradually become apparent that structural damage alone cannot satisfactorily explain age-related functional losses; therefore, even an increase in antioxidant defences cannot reverse the ageing process [160]. Antioxidants, including the Trx system, can mitigate oxidative damage, but this may not prolong a given individual’s life [161]. The disruption of either Trx-2 or Trx-2 genes strongly reduces the lifespan of the long-lived mitochondrial worm mutants, indicating that the mitochondrial Trx system is strictly required for worm longevity [162]. The validity of this hypothesis for insects was undermined by findings in the honey bee, which is a suitable model organism for ageing due to the different lifespans of the different castes. An analysis of the capacity of the antioxidant system of old honey bee queens indicated that longevity was not associated with higher expression of antioxidant genes, including the TrxR-1 gene [163]. The longevity of queens is thus not achieved by enhanced antioxidant protection but by a different mechanism. In bee queens and in winter long-lived bee workers, it is most likely a combination of multiple factors such as higher telomerase activity in queens, diet composition, higher vitellogenin levels and overall increased antimicrobial activity [164,165,166,167].

The transcriptomic study investigated the influence of pollen nutrients on the transcriptome of worker bees parasitised by the mite *Varroa destructor*, known for suppressing bee immunity and decreasing lifespan [168]. Sod and Trxr-1 were upregulated in nonparasitised pollen-fed bees, whereas pollen supplementation did not affect Trx-r1 expression in *Varroa*-infected bees. This suggests that SOD and TrxR might be involved in the pollen-induced molecular mechanisms that enhance the life expectancy of honey bees. Selenium supplementation significantly upregulated the four major honey bee antioxidant genes (Sod1, Trxr1, MsrA, Cat) in caged bees at low, tolerated and higher sublethal doses of selenium [169]. Interestingly, in a follow-up study aimed to analyse the effects of pesticide treatment, caging bees activated the upregulation of the gene group involved in the antioxidative process when compared with hive bees [170]. During a 3-week treatment period, genes belonging to the thioredoxin family, such as Trx-1, Trx1-like2 and Trxr-1, as well as the glutaredoxin and glutathione families, were induced in parallel to higher protein carbonylation in caged bees. Thus, the caging stress triggered the bee antioxidant response to mitigate this stress and prevent further cellular damage. The effects of bee treatment can be, to a large extent, influenced by the type, doses and application method of compounds tested on caged or hive bees. In newly emerged bee workers, imidacloprid induced a strong upregulation of TrxR-1 and CAT with a lower rate of midgut apoptosis, whereas the application of the imidacloprid and coumaphos mixture caused the downregulation of antioxidant genes with noticeable midgut tissue damage [171].

### 4.3. Regulation of Nitrosative Stress

In addition to its function as a regulator in redox processes associated with oxidative stress, the Trx system also plays a role in the regulation of nitrosative stress. Nitric oxide (NO) reversibly binds to the thiol group in the Cys side chain in a reaction called S-nitrosylation. This post-translational modification regulates many molecular functions, including enzyme activity, translocation, protein–protein interactions and degradation [127,128]. In mammalian Trx1, Cys conserved outside the protein’s active site is first nitrosylated and then NO is transferred from Trx to the Cys of other proteins by transnitrosylation [126]. Trx can also denitrosylate nitrosylated proteins back to their original state by simultaneously forming reduced NO, nitroxyl [71,172]. Nevertheless, the potential involvement of the insect Trx system in the control of protein S-nitrosylation has not been studied so far.

### 4.4. Redox Signalling, Cell Death and Apoptosis in Insects

Thioredoxins maintain a reduced environment inside the cell by reducing protein disulphides, mediating the cellular response to changes in the redox state. Thus, Trx may function as a redox regulator of signalling molecules and transcription factors [78]. The function of Trx as a regulator of cell death has been described for mammalian cytosolic Trx1. Reduced Trx1 binds to apoptosis signal-regulating kinase 1 (ASK1), also referred to as mitogen-activated protein kinase 5, causing the inhibition of the kinase activity. Moreover, by forming a Trx1-ASK1 complex, ASK1 is targeted for ubiquitination. The oxidation of Trx1 results in its dissociation and the activation of ASK1, which is part of the mitogen-activated protein kinase (MAPK) pathway. This further activates Jun N-terminal kinase (JNK) and p38 MAPK, which are responsible for apoptosis induced by oxidative stress [78]. Conversely, to induce apoptosis in the cell, Trx1 activity must be inhibited. Due to oxidative stress, Trx1 can be oxidised and dissociate, restoring ASK1 activity [73]. A Trx1-interacting protein (Txnip) also serves to regulate Trx1 by binding to the active site of reduced Trx1 and forming a mixed disulphide with it, thereby inhibiting it and allowing the activation of the MAPK pathway, leading to apoptosis [134]. The activity of the phosphatase and tensin homolog (PTEN) is also crucial for the proper regulation of apoptosis. PTEN can be activated by the reduction of the disulphide bond by the action of Trx. Active PTEN inhibits the Akt signalling pathway, which is involved in cell growth and proliferation; in contrast, apoptosis is stimulated when the Akt pathway is inhibited [73]. Trx1 also mediates the activation of several transcription factors involved in cell growth, apoptosis and inflammation. These transcription factors contain redox-sensitive cysteines in their DNA binding domain, such as NF κB (nuclear factor kappa B), AP-1 (activator protein) or Ref-1 (redox factor 1) [69].

The apoptosis-inducing factors (AIF) are a family of moonlighting proteins with poorly characterised oxidoreductase activity and the capability to induce programmed cell death [173]. AIFs have been proposed to serve as conserved redox switches that detect metabolic conditions on the mitochondrial surface and translate them into binary life-or-death outcomes [174]. The characterisation of the Trx-AIF interaction uncovered the role of reduced Trx1 in regulating AIF-dependent cell death by avoiding AIF-mediated DNA damage [175]. Under physiological conditions, cytosolic Trx1 interacts with the apoptosis induction factor (AIF), with the cysteines in the Trx active site playing a crucial role. Under oxidative stress, Trx-AIF interaction is disrupted, followed by the nuclear translocation of AIF.

In an initial study of AIFs in insects, the knockout of zygotic *D. melanogaster* AIF (DmAIF) caused decreased embryonic cell death and the persistence of differentiated neuronal cells at late embryonic stages [176]. The results confirmed the presumed AIF as a mitochondrial effector regulating both normal mitochondrial function and caspase-independent cell death. In *A. cerana*, AccAIF3 transcripts were induced by cold, CdCl_2_, HgCl_2_, UV, pyriproxyfen and cyhalothrin but were downregulated by ecdysone [177]. The expression of AIF3 in *A. cerana cerana* and *A. mellifera ligustica* also responded to infection by a fungal pathogen *Ascosphaera apis* in a species-specific manner, suggesting that AIF3 may play a vital role in the response to both abiotic and biotic stresses. Interestingly, the predicted transcript of AmAIF3 in the GenBank database (XM_625032.6) was originally deposited as a predicted transcript of *A. mellifera* thioredoxin reductase 3 (AmTrxR3) gene (XM_625032.4). Collectively, the function of AIF within the thioredoxin system in the control of redox homeostasis in insect cells requires further investigation.

### 4.5. Regulation of Cytoskeleton Structure

The Trx system can regulate the components that make up the cytoskeleton of eukaryotic cells. The oxidation of β- and γ-actin is known to interfere with the polymerisation process of actin, which cannot be synthesised. However, the Trx system allows the oxidation to be reversed and the actin formation to be restarted [131]. Also, the oxidation of the neuron-specific microtubule-associated protein MAP2 and tau inhibits microtubule formation from tubulin, but thanks to the Trx system, MAP disulphides can be reduced [178]. Redox mechanisms could regulate the polymerisation of tubulin in a Trx-dependent manner, as the Trx/TrxR system was observed to be able to reduce one intra-subunit disulphide bond in the tubulin dimer, leading to a partially inhibited microtubule assembly [179]. These observations point to the poorly understood complexity with which the Trx system may be involved in modulating cytoskeletal structure and shape cell dynamics.

## 5. Thioredoxin System in Honey Bees, the Model Social Insect

Honey bees are important insect pollinators of wild plants and economic crops and play a key role in the ecosystem. Various biotic and abiotic factors threaten the survival of bee communities, and it is therefore necessary to find ways to increase the resilience of honey bee colonies to stress conditions [180,181,182]. As flying insects, bees are constantly exposed to stressful conditions when collecting pollen and nectar. The metabolic rate of the flying bees increases 10 to 100 times during flight compared to non-flying workers. Therefore, an efficient and robust antioxidant defence is needed to maintain homeostasis and prevent oxidative stress [50,55]. The antioxidant system of bees contains all components of the thioredoxin system. Three genes encoding three Trx variants were found in the genome of *A. mellifera*: AmTrx-1 with putative mitochondrial localisation, AmTrx-2 and AmTrx-3. There is only one gene for TrxR encoding two variants of the enzyme, AmTrxR-1 and AmTrx-2, but neither has been confirmed to localise in the mitochondria. However, it has been suggested that one of the TrxR variants should be targeted to mitochondria to provide reduced Trx for mitochondrial TPx crucial for regulating H_2_O_2_ levels, as catalases are absent in bee mitochondria [21]. The AmTrxR gene is 1674 bp long, and the resulting AmTrxR-1 enzyme is composed of 491 AMKs, whereas in the related species *A. cerana*, the AcTrxR gene length is 1620 bp, and the enzyme consists of 494 AMK [151].

Bees are exposed to various biotic and abiotic factors that can induce oxidative stress and the subsequent activation of antioxidant defences involving the thioredoxin system. Biotic stressors include viruses, bacteria, fungi and parasites, whereas the main abiotic factors include changes in environmental temperature, pesticide exposure and nutrition [181]. An increased expression of the TrxR1 gene has been reported in *A. mellifera* and *A. cerana* following exposure to high (37 °C) or low (4 °C) temperatures and after the injection of the oxidants paraquat or H_2_O_2_ [151]. Exposure to UV light or elevated temperature (37 °C) significantly increased the expression of the TrxR gene in all tissues in *A. cerana* [152]. An increased expression of the AccTrx 2 gene was also found in *A. cerana* under conditions of varying ambient temperatures (4, 16 and 25 °C) and following the injection of oxidative stress-inducing compounds H_2_O_2_, HgCl_2_ and pesticides cyhalothrin, paraquat and phoxim [83]. An analysis of AccTrx-1 gene expression provided the same results, except for reduced expression after exposure to HgCl_2_ [141]. Koo et al. [151] analysed the expression of the TrxR gene in two bee species, *A. mellifera* and *A. cerana*. They found that it increased after exposure to low or high temperatures (4 °C and 37 °C) and treatment with the oxidants paraquat and H_2_O_2_. In addition, *A. cerana* appeared to be more resistant to low-temperature stress, as the gene expression for TrxR did not reach the same levels as in *A. mellifera*, and the bees appeared visibly calmer.

Female sperm storage is a phenomenon in which females store sperm in a specialised organ for periods lasting from a few hours to several years, depending on the species. Eusocial hymenopterans (ants, social bees and social wasps) hold the record for sperm storage duration [183]. Honey bee queens have the ability to store and maintain drones’ sperm in special organs called spermatheca. The metabolic activity of sperm in this organ is vital for prolonged sperm utilisation; however, ROS are produced during aerobic metabolism, which can reduce sperm viability. Queen spermathecae and the seminal fluid of drones contain antioxidant enzymes with increased activity to maintain sperm viability, including the components of the thioredoxin system Trx-2 and TrxR-1 [184,185,186]. Interestingly, significant differences in the amount of antioxidants between virgin and mated queens have been observed. A higher expression of Trx-2 and TrxR-1 genes was found in fertilised queens compared to virgin queens. The factors leading to the activation of these genes may originate in the semen, seminal fluid or the queen’s reproductive organs when she begins to produce and lay eggs [187]. The authors monitored the expression of several antioxidant protein genes, including TrxR and Trx, in honey bee queen spermathecae. In fertilised queens, the expression of TrxR and Trx was up to twice as high as in unfertilised queens, indicating a possible function of the Trx system in protecting honey bee sperm from ROS degradation. In a subsequent study, the obtained RNA sequencing data confirmed a three-fold higher expression of the TrxR-1 gene in the mated queens compared to virgin queen spermatheca [188].

Recent advances in understanding the roles of the Trx system in the interaction of honey bees with intestinal symbionts and pathogens can pave the way to design novel and efficient treatments against *Nosema ceranae*, an invasive microsporidia affecting honey bee health worldwide. Bacterial microbiota are well known for their crucial role in supporting bee growth with respect to nutrition and gut immune protection [189]. Maximising the capabilities of native gut symbionts has the potential to maintain the health status and promote the growth and vitality of bees [190,191,192]. Bee gut-specific bacterial species *Snodgrassella alvi* was found to inhibit microsporidia proliferation, likely by the induction of bee immune response based on increased ROS in the gut epithelium [193]. Interestingly, this study proves that *N. ceranae* exploits the thioredoxin and glutathione systems to defend against oxidative stress caused by increased ROS levels in bee guts. The RNAi knockdown of thioredoxin reductase and the GSH synthetising enzyme γ-glutamyl-cysteine synthetase reduced the *Nosema* spore load in infected bee guts. The pathogen thioredoxin system is thus necessary to maintain a balanced redox state essential for a successful infection.

## 6. Conclusions and Future Directions

Vast and detailed knowledge of the Trx system in diverse organisms has accumulated during the past 50 years since the first description of thioredoxin in *E. coli* [74]. Currently, many aspects of their structure and reaction mechanisms involved in their catalytic and redox activities are well understood, including their compartment-specific functions and interacting partners. Notably, the advances in the mammalian Trx system have been paralleled by similar progress and detailed descriptions of the Trx system in model insect species, namely *D. melanogaster* and *A. gambiae*.

Nevertheless, significant gaps still exist in our knowledge of the precise regulation of the Trx system within redox niches of intracellular compartments in connection with other components of the redox regulatory network. Furthermore, there is relatively limited knowledge of the interacting partners and target protein in insect cells compared to other organisms. Recent advances brought the implementation of new proteomic tools to characterise the Trx interactome in bacteria, plants and human cells [194,195,196]. It can be envisaged that applying these tools in insect studies would be feasible to provide more information to understand Trx function in insect development and stress responses.

Genetically encoded redox biosensors have emerged as highly valued tools for the analysis of redox mechanisms at a subcellular resolution, as they can report the localisation and redox state of the target analyte based on changes in the fluorescence signal intensity [197]. The application of genetically encoded biosensors for analysing the glutathione redox potential in mitochondria revealed extensive crosstalk between the mitochondrial glutathione and thioredoxin systems and their decisive involvement in removing H_2_O_2_ produced in mitochondria [198]. The biosensor HyPer1, developed originally as an H_2_O_2_ probe, was utilised to obtain a deeper insight into the thioredoxin- and GSH-dependent reductive activity in cellular compartments [199]. Interestingly, the data suggest that Trx/TrxR predominantly reduces HyPer1 in the cytosol and nucleus, whereas GSH mainly reduces it in the mitochondria. As repeatedly described in this review, the efficient function of both Trx- and GSH-dependent redox control mechanisms are decisively dependent on the cellular sources of reduced cofactor NADPH as the source of reductive power. It is evident that new tools enabling the assessment of the NADP(H) redox status will be highly valuable. Recently, a novel genetically encoded ratiometric biosensor NERNST was reported to selectively monitor the NADP(H) redox potential in bacterial, plant and animal cells and organelles such as chloroplasts and mitochondria [200]. The adaptation and implementation of these molecular tools to study Trx and other redox systems in insect cell lines are yet to occur.

Collectively, the accumulation of the recently achieved progress and newly available tools to study the Trx system within the context of the precise spatiotemporal redox regulations shows promise in advancing our understanding of the Trx system in insects. Future studies on its regulation and targets will uncover the emerging role of the Trx system beyond the antioxidant system in model insect species. Importantly, these advancements can be exploited to investigate the Trx system further in agronomically or ecologically relevant insect species, such as honey bees. Basic research on Trx and TrxR functions can be applied to novel concepts to increase bee health and their resistance to stress conditions induced by malnutrition, pathogens and environmental pollutants.

## Figures and Tables

**Figure 1 insects-15-00797-f001:**
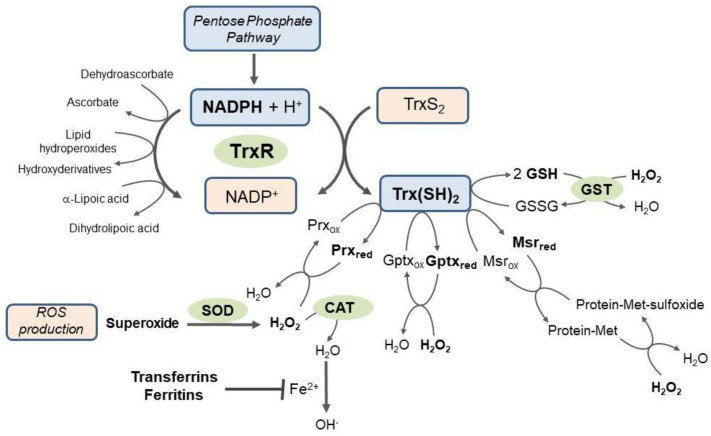
The overview of the roles of the thioredoxin system within insect antioxidant mechanisms. CAT, catalase; Gptx, glutathione peroxidase-like proteins; GSH, reduced glutathione; GSSG, oxidised glutathione; GST, glutathione transferases; Mrs, methionine sulphoxide reductases; Prx, peroxiredoxins/thioredoxin peroxidases; SOD, superoxide dismutases; Trx, thioredoxin; TrxR, thioredoxin reductase.

**Figure 2 insects-15-00797-f002:**
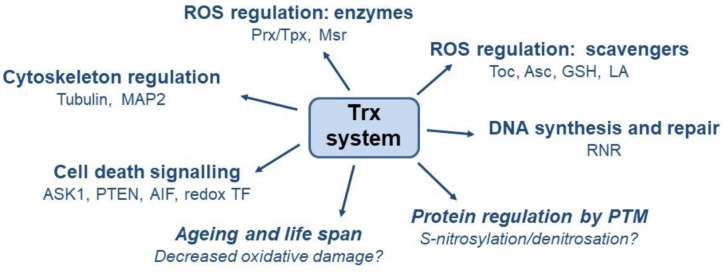
Functions of the insect thioredoxin system. AIF, apoptosis-inducing factors; Asc, ascorbate; ASK1, apoptosis signal-regulating kinase 1; GSH, reduced glutathione; LA, lipoic acid/lipoamide; MAP2, microtubule-associated protein 2; Mrs, methionine sulphoxide reductases; Prx/Tpx, peroxiredoxins/thioredoxin peroxidases; PTEN, phosphatase and tensin homolog; RNR, ribonucleotide reductase; TF, transcription factors; Toc, tocopherols; Trx, thioredoxin.

## Data Availability

No new data were created or analysed in this study. Data sharing is not applicable to this article.

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
