# Peer review of "Thioredoxin System in Insects: Uncovering the Roles of Thioredoxins and Thioredoxin Reductase beyond the Antioxidant Defences"

_insects, 2024, doi:10.3390/insects15100797_

Round 1

Reviewer 1 Report

Comments and Suggestions for Authors

"Thioredoxin system in insects: uncovering the roles of thioredoxins and thioredoxin reductase beyond the antioxidant defences" is a complete, well organized and well written review on antioxidant systems and related antooxidant compouds in insects.  Yes, this review is important to provide a comprehensive report on the topic, useful for the reader not to go trough literature to have the overview. 

Only minor remarks, detailed below, require attention, to make the paper suitable for publication.

  Lines 230, 234, ound brackets are detected, references/tex-t? are missing? 

The "presentation" of the work in the Introduction (lines 84-92) is appreciated. However, and not mandatory, given that this is a big, articulated text, a schematic index could be useful to the reader.

Aware that gaps still exist, basic research on Trx and TrxR functions is proposed to be applied to novel concepts to increase bee health and resistance to stress conditions induced by malnutrition, pathogens, and environmental pollutants.

Comments on the Quality of English Language

good

Author Response

Comments 1: Thioredoxin system in insects: uncovering the roles of thioredoxins and thioredoxin reductase beyond the antioxidant defences" is a complete, well organized and well written review on antioxidant systems and related antoxidant compouds in insects.  Yes, this review is important to provide a comprehensive report on the topic, useful for the reader not to go trough literature to have the overview. Only minor remarks, detailed below, require attention, to make the paper suitable for publication.

Reply 1: We thank the reviewer for his positive evaluation of our manuscript.

Comments 2: Lines 230, 234, round brackets are detected, references/text? are missing? 

Reply 2: The round brackets at these places were deleted from the text.

Comments 3: The "presentation" of the work in the Introduction (lines 84-92) is appreciated. However, and not mandatory, given that this is a big, articulated text, a schematic index could be useful to the reader.

Reply 3: Thank you for your suggestions; in fact, we were considering to include an index to our manuscript, but as far as we understand, indexes are not used within the MDPI manuscript template style. 

Reviewer 2 Report

Comments and Suggestions for Authors

The review article, "Thioredoxin system in insects: uncovering the roles of thioredoxins and thioredoxin reductase beyond the antioxidant defences", provides a detailed summary of the Thioredoxin system in insects with a specific focus on the Honey bee.

The introduction gives a comprehensive overview of the importance of the system in organisms. However, it is too detailed and jumps/ lost transition from ROS/RNS to Trx systems. The introduction also mentions some content covered in section 4 (function of Trx system).

The opening sentence of the introduction and the opening sentence of section 2.1 are repetitive. Lines 71-73 and Lines 123-124 are the same. In many sections, the role of ROS/RNS is mentioned. Please avoid redundancy in describing the role and focus more on the specific aspects of the insect system, as the title suggests. Of interest to the field right now is how NAD/NADH ratios regulate redox stress. The article completely misses this. How does this affect Trx in insects? Also, extensive discussion of Glutathione distracts from the Trx focus.

Section 3.2 has two subheadings, which overlap and can be combined as structure and mechanism. Also, section 3.3 has three subheadings, and the 3.3.2 title is the same as the section heading.  Please consider consolidating them. Again, Section 4 has a lot of subsections 4.1-4.5, which are overwhelming and segmented. Please consider consolidating the sections for logical flow and better clarity.

As a reader, I needed to remember the logical sequence after reading a couple of pages. Interestingly, the review cited 200 references, but I did not gain that much knowledge from reading this since it is too complicated to understand. Overall, this review article is very descriptive and redundant and contains information that distracts from the paper's goal, with no tables and figures.

Comments on the Quality of English Language

Nothing specific

Author Response

Comments 1: The review article, "Thioredoxin system in insects: uncovering the roles of thioredoxins and thioredoxin reductase beyond the antioxidant defences", provides a detailed summary of the Thioredoxin system in insects with a specific focus on the Honey bee. The introduction gives a comprehensive overview of the importance of the system in organisms. However, it is too detailed and jumps/ lost transition from ROS/RNS to Trx systems. The introduction also mentions some content covered in section 4 (function of Trx system).The opening sentence of the introduction and the opening sentence of section 2.1 are repetitive. Lines 71-73 and Lines 123-124 are the same. In many sections, the role of ROS/RNS is mentioned. Please avoid redundancy in describing the role and focus more on the specific aspects of the insect system, as the title suggests.

Reply 1: We appreciate this reviewer's comment. We have revised the introductory sections 1 and 2 to remove redundancy with other parts of the manuscript. We have also intended to shorten section 2 and focus it more on the thioredoxin-related components of the antioxidant system.

Comments 2: Of interest to the field right now is how NAD/NADH ratios regulate redox stress. The article completely misses this. How does this affect Trx in insects?

Reply 2: We appreciate this inspiring comment. As far as we could find in the available literature, the only study dealing with NAD+/NADH ratio in insects was published by Hansford (1975), which reported that NADH increased and the NAD+/NADH ratio decreased during flight in blowfly flight muscles (DOI: 10.1042/bj1460537). In general, the connections between NAD+ metabolism and NADPH/NADP+ ratio are still poorly understood even in other model organisms. We are not aware of any study addressing this issue in insect species; for this reason, this putative aspect of redox regulation has not been covered in our review paper.

Comments 3: Also, extensive discussion of Glutathione distracts from the Trx focus.

Reply 3: We appreciate this reviewer's comment. We considered it appropriate to cover the regeneration of reduced glutathione in detail, as in insects, this is provided by the Trx system due to the absence of glutathione reductase. However, we have revised the manuscript to shorten the discussion of glutathione-related topics whenever possible.

Comments 4: Section 3.2 has two subheadings, which overlap and can be combined as structure and mechanism. Also, section 3.3 has three subheadings, and the 3.3.2 title is the same as the section heading.  Please consider consolidating them.

Reply 4: Section 3.3.2 has been renamed to "Specific features of insect thioredoxin reductases" as suggested.

Comments 5: Again, Section 4 has a lot of subsections 4.1-4.5, which are overwhelming and segmented. Please consider consolidating the sections for logical flow and better clarity.

Reply 5: We have revised Section 4 and corresponding subsections to consolidate their content and decrease segmentation.

Comments 6: As a reader, I needed to remember the logical sequence after reading a couple of pages. Interestingly, the review cited 200 references, but I did not gain that much knowledge from reading this since it is too complicated to understand. Overall, this review article is very descriptive and redundant and contains information that distracts from the paper's goal, with no tables and figures.

Reply 6: We have included two figures to illustrate the main ideas and summarise the paper's content.

Comments on the Quality of English Language: Nothing specific

Reviewer 3 Report

Comments and Suggestions for Authors

Authors present a critical and up-to date review on antioxidant defense systems in insects with a focus on the thioredoxin system. This is of high interest for entomologists and biologists in general, and also of practical importance for studying agroeconomically important and eusocial insects.

The manuscript needs some editorial and lingual improvement (see below).

Comments on the Quality of English Language

English is mainly fine, but some grammatical and typing errors must be corrected.

Author Response

Comments 1: Authors present a critical and up-to date review on antioxidant defense systems in insects with a focus on the thioredoxin system. This is of high interest for entomologists and biologists in general, and also of practical importance for studying agroeconomically important and eusocial insects. The manuscript needs some editorial and lingual improvement (comments in attached PDF file).

Reply 1: We thank the reviewer for evaluating our manuscript. All corrections marked in the attached PDF file have been done in the manuscript.

Comments 2: One or two illustrations would certainly upgrade the review

Reply 2: As suggested, we have included new Figures illustrating the connection of the Trx system within insect antioxidant defence and summarising the functions of the Trx system in the regulation of other enzymes and metabolic pathways.

Comments 3: on the Quality of English Language: English is mainly fine, but some grammatical and typing errors must be corrected.

Reply 3: We have performed a thorough check of the entire manuscript to correct grammatical and typing errors as requested.

Reviewer 4 Report

Comments and Suggestions for Authors

Respected Authors,

Please find the comments and suggestions for your manuscript in the attached Word file.

Comments on the Quality of English Language

The manuscript is written fairly well and coherently. However, it would benefit greatly from restructuring run-on sentences and using more punctuation marks such as commas. This latter would improve the general flow of the text, giving the reader more time to digest the complex content that is presented therein.

Author Response

Comments 1: In this review, the Authors have offered a comprehensive look at the current state of knowledge regarding the thioredoxin-based antioxidative defence system in insects, which is underreported. They go over the fundamental characteristics of this antioxidative defence system, the classification of its components, different functions in maintaining organism homeostasis, interplay and interconnection with other major biomolecules, as well as the specific case of the thioredoxin system in the model insect Apis mellifera.

Reply 1: We thank the reviewer for his evaluation and suggestions for improving the manuscript.

Comments 2: Apart from this, the Authors also introduce a brief overview of the general antioxidant defence system in insects. However, this segment of the manuscript detracts too much from the main topic, which is the thioredoxin system, as it goes into deep detail regarding antioxidant defence system components that lack meaningful connection with the thioredoxin system. This segment, mainly contained in the heading 2. Thioredoxin system within the insect antioxidant regulation and defence, should be revised and shortened, in order to be better focused on the actual thioredoxin system. Additionally, considering the contents of this heading, it is advised to rename it to Brief overview of insect antioxidant system. Authors are also recommended to create a figure which would illustrate the connection of the thioredoxin system with all these other antioxidant defence system components.

Reply 2: We appreciate this reviewer's comment. The mentioned MS segment has been renamed as suggested to "Brief overview of insect antioxidant system". We have also intended to shorten this segment and focus it more on the thioredoxin system. We have also included a new Figure illustrating the connection of the Trx system with insect antioxidant defence.

Comments 3: Apart from this, the Authors are advised to thoroughly proofread the manuscript, as there are many spelling and grammatical errors throughout the text. For example, Latin names of species are not italicized in some instances (Line 88), incorrect words are used (Line 140, "of" instead of "or"), hydrogen peroxide is incorrectly annotated (Line 154), leftover parentheses (Lines 230 and 234), species are incorrectly named (Line 235, Anopheles aegypti instead of Aedes aegypti), references are incorrectly formatted in-text (Line 701). When a species is mentioned in the text for the first time, its Latin name should be written out completely. Whenever that species is mentioned in the text again, the Latin name should be shortened.

Reply 3: We have extensively proofread the entire manuscript to eliminate the mentioned errors. We have checked particularly the correct usage of Latin names.

Comments 4: Also, the Authors should review MDPI's instructions on how to present references in the reference list, as many of the references are not formatted according to MDPI's style.

Reply 4: We have corrected the presentation of the references and the formatting of the reference list according to the MDPI style, as requested.

Comments 5: Comments on the Quality of English Language

The manuscript is written fairly well and coherently. However, it would benefit greatly from restructuring run-on sentences and using more punctuation marks such as commas. This latter would improve the general flow of the text, giving the reader more time to digest the complex content that is presented therein.

Round 2

Reviewer 2 Report

Comments and Suggestions for Authors

The authors responded to the comments and included figures. The revised version is informative.

Author Response

Comments 1: The authors responded to the comments and included figures. The revised version is informative.

Reply 1: We thank the Reviewers for their valuable comments and suggestions.

Reviewer 4 Report

Comments and Suggestions for Authors

Dear Authors,

Thank you for taking into consideration the comments and suggestions that were made regarding your manuscript. It has been substantially improved, and the discussion benefits greatly from the included figure.

Having read the revised version of your paper, I would recommend another round of proofreading, as some minor mistakes have been overlooked and should be remedied. For example, the chemical formula for hydrogen peroxide in Lines 143, 414, 418, 539 is not formatted correctly, the abbreviation for Ribonucleotide reductase is incorrectly written as RNA (Line 740) and RNK (Line 745).

Author Response

Comments 1:  Dear Authors, Thank you for taking into consideration the comments and suggestions that were made regarding your manuscript. It has been substantially improved, and the discussion benefits greatly from the included figure. Having read the revised version of your paper, I would recommend another round of proofreading, as some minor mistakes have been overlooked and should be remedied. For example, the chemical formula for hydrogen peroxide in Lines 143, 414, 418, 539 is not formatted correctly, the abbreviation for Ribonucleotide reductase is incorrectly written as RNA (Line 740) and RNK (Line 745).

Reply 1: We thank the Reviewer for his valuable comments and suggestions. As required, we have performed another round of manuscript proofreading to correct the mentioned mistakes and other mistakes in the text.